# Diamond Open Access in Norway 2017–2020

**Jan Erik Frantsvåg** 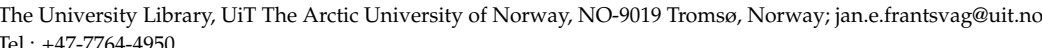

The University Library, UiT The Arctic University of Norway, NO-9019 Tromsø, Norway; jan.e.frantsvag@uit.no; Tel.: +47-7764-4950

**Abstract:** We see from information published elsewhere that Gold OA is on the increase globally. The OA Diamond study indicates that Diamond OA is an important component of scholarly communications, with an estimated 8–9% of the total global scholarly output. These numbers, however, are on a global scale and are not necessarily representative of any given country; country case studies are needed to find this information. Norway is a country where the government has declared a 100% OA goal and most research has public funding. Norway has good financing structures for various models of OA, and it has a national CRIS system. This study tries to find and present numbers for articles in scholarly journals to describe both recent developments and relative numbers for Norway as a whole, and for scholarly fields in Norway, with regards to Diamond OA. Numbers for and development of Gold OA will also be given and commented upon to some extent.

**Keywords:** Diamond OA; Norway; Gold OA; scholarly publishing

## 1. Introduction

There are two basic models of open access (OA): Gold OA, which is OA on the publisher's side, and Green OA, which is OA on the author's side. Content that is not OA, i.e., content you have to pay to get access to, is toll access (TA) [1].

Hybrid is Gold OA in publications that are TA, but which allow part of the content to be "set free", generally for a fee [1]. Another model that will be mentioned is Bronze, which is content "made free-to-read on the publisher website, without an explicit Open license" [2]. Bronze is neither OA nor TA, but something in-between, and not necessarily a stable categorization—publishers can bar access whenever they want and make content TA or inaccessible.

Statistics for the following three models are studied, with an emphasis on the last model, though numbers for the other two models will be given for comparison and to show the relative frequency of Diamond OA:

- Toll access (TA): Readers must pay to access content, authors normally do not pay for publishing per se, though some payment may be expected to be made for overlength articles, high numbers of tables or illustrations, color etc. Hybrid open access (open access articles published in TA journals), which is Gold OA, cannot be quantified in this study, where data are at the journal and not the article level. Therefore, in this study, Hybrid will be classified as TA.
- Article processing charge (APC)-based OA (OA-APC): Free to access for readers, but authors must ensure financial contributions to the journal in order to publish. This is generally funding from the author's institution, though not necessarily.
- Diamond OA: Neither readers nor authors must pay to access or to publish content. Costs are borne by third parties. (Definitions of Diamond OA vary, see e.g., [3], the definition used here conforms to the one used in [4]).

Norway has a centralized CRIS (Cristin, The Current Research Information System in Norway, https://www.cristin.no/, accessed on 1 February 2022) where all Norwegian research from the higher education (HE) sector, health sector, and institute sector is registered [5]. (Cristin was rolled out on a large scale in 2011 after an earlier gradual start, but

more organizations have been added to the list of reporting institutions since [6]). Only private research is systematically not registered, in addition to some research performed by governmental organizations that are not regarded as research organizations, and which do not perform much research. The completeness of registrations and the quality of information is believed to be nearly 100% for the organizations involved because publishing activities are monitored and for most organizations (e.g., all HE institutions), they also affect the organization's funding. As this study aims to look at the development of Diamond publishing over time, and distribution over scholarly fields, any lacking or imprecise information will have no important consequences for the results of the study.

Only content in journals and serials with an ISSN number is studied. This is both due to the fact that policies and funding mechanisms for books, book chapters, etc. are not well developed yet, and because some information is not so readily available, making it harder to study. In addition, articles in journals are the bulk of scholarly outputs in Norway, though books have an important place in some fields, especially in humanities and social sciences (HSS).

Green open access (OA at the author's side, generally through self-archiving in repositories), while important, is not a part of this study. It should be noted, however, that the national CRIS is an important tool when working to collect manuscript versions for self-archiving, and that is has mechanisms making depositing relatively easy for authors. In Norway, nearly all institutions have an institutional repository; some depositing is also done in international subject-based repositories.

*Earlier Studies*

A global study on Diamond OA is the "OA Diamond Study" [4] which was commissioned by Science Europe for cOAlition S. This is a global study, giving numbers for Diamond OA journals and publishing volumes in these journals. The study is based partly on DOAJ journal metadata, partly on a survey with a global outreach. The OA Diamond Study estimates that 8–9% of the global scholarly output is published in Diamond OA journals. This must be considered a rough estimate, given that it is very difficult to find really reliable numbers for, e.g., total volume of scholarly articles. The study also shows that Diamond is much more important in HSS than in science and medicine. Another finding is that Diamond OA journals on average publish fewer articles than APC-based OA journals. This corresponds well with the findings in [7] that Diamond OA journals are 69% of the journals in DOAJ, but publish only 35% of the articles in the journals indexed in DOAJ in 2020. The geographic distribution of Diamond versus APC-based OA journals is also very diverse, with nearly all Latin American OA journals being Diamond, compared to a little over half of African and western European OA journals. Unfortunately, the survey only had one respondent from Norway (in addition to one from Iceland, two from Sweden, and five from Finland), so the survey which collected information on many aspects not covered by the DOAJ journal metadata cannot inform this study.

Other studies generally look at number of journals belonging to a specific country or countries, not at publishing output (articles) from these countries. They cannot be compared with the results of this study where the Norwegian output is studied, not Norwegian journals per se. A good reason for the lack of such studies is the lack of relevant (CRIS) data on the national level—Scopus- or Web of Science-based studies are skewed towards TA and APC-based publishing, as Diamond journals are underrepresented in their data.

## 2. Materials and Methods

On 31st May 2021, a journal metadata file was downloaded from DOAJ.

On 2nd June 2021, a data file was received from The Norwegian Directorate for Higher Education and Skills (HK-dir). (All files are made available in UiT Open Research Data [8]). The data were harvested from Cristin for the years 2017–2020. In the file, there is one line per year per journal in which any scholarly content with an author from an institution

using Cristin was published. Journal title, whether in DOAJ or not, how many articles were published, and the sum of article fractions (see below) for Norwegian authors reporting to Cristin, scholarly field and sub-field, ISSNs, etc. is information found in this file. The information does not specify which scholarly sector (higher education, health, research institute, etc.) the authors belong to, the data are given for Norway as a whole. As this information is journal-based, hybrid OA cannot be discovered, only OA in OA journals listed in DOAJ (Directory of Open Access Journals, considered the authoritative database of OA journals) can be identified as OA.

This is, of course, a source of error, as there are numerous journals that are OA but are not registered in DOAJ for some reason. OA will be underestimated when a DOAJ listing is used as the definition of OA, but there are few, if any, viable alternatives to using DOAJ as the definition of which journals are OA. Furthermore, many of those journals missing from DOAJ will be Diamond OA. Ref. [9] estimates that less than 50% of Nordic OA journals are listed in DOAJ. The Polish database Arianta https://arianta.pl/ (accessed on 3 January 2022) indicates the existence of more than 1600 Polish scholarly OA journals, compared to 786 Polish journals listed in DOAJ. Another study [10] indicates that more than 2000 bona fide OA journals may have been removed during DOAJ's re-accreditation process in 2016. Consequently, our numbers of OA and Diamond OA journals, and their share of output, will be conservative estimates and TA will be correspondingly overestimated. However, I see no reason to believe this problem has changed much over the years studied, so it should not affect our picture of how various form of access have developed recently.

Based on ISSNs in the file from HK-dir, information about APCs was retrieved from the DOAJ metadata file to supplement information in the file from HK-dir. The journals in the HK-dir file were then classified:

(1)　If not in DOAJ: toll access (TA).
(2)　If in DOAJ, but year added to DOAJ higher than the reporting year: TA.
(3)　If in DOAJ and year added to DOAJ lower or equal to reporting year, while APC = "Yes": OA-APC.
(4)　If in DOAJ and year added to DOAJ lower or equal to reporting year, while APC = "No": OA Diamond.

All publishing was thus assigned one of the three categories TA, OA-APC, or OA Diamond. The comparison of reporting year and year added to DOAJ means a journal may change from being TA to one of the other categories from one year to another. As there may be a time lag between actually becoming OA and being listed in DOAJ, the shift from TA to OA may have happened earlier than the data indicate.

When counting, the sum of article fractions was used, where each of $n$ authors of an article is assigned $1/n$ of the article, instead of counting every article with even a minimum of Norwegian author involvement as a whole article. One important reason to do this is to ensure that numbers for fields where multi-author articles, often with a large number of authors and widespread international cooperation, are not inflated when compared with fields where articles with single or few authors and less international cooperation is the norm. Moreover, when comparing studies across data sets from different countries, counting articles would inflate numbers even more—when counting articles, one needs to eliminate articles present in more than one set of data. Summing article fractions will eliminate that problem.

Looking at development over time, the complete data over the years 2017–2020 were used. Looking closely at how the different models are distributed over scholarly fields and sub-fields, only data for the last two years, 2019–2020, were used—it is the current situation that is the most interesting, older data can distort this view of the current situation.

## 3. Results

### 3.1. The General Picture

Figure 1 illustrates the total of Norwegian scholarly production over all sectors and all fields. It should be easy to see that there is a general tendency over time for the TA share to become smaller, while OA-APC and OA Diamond grows.

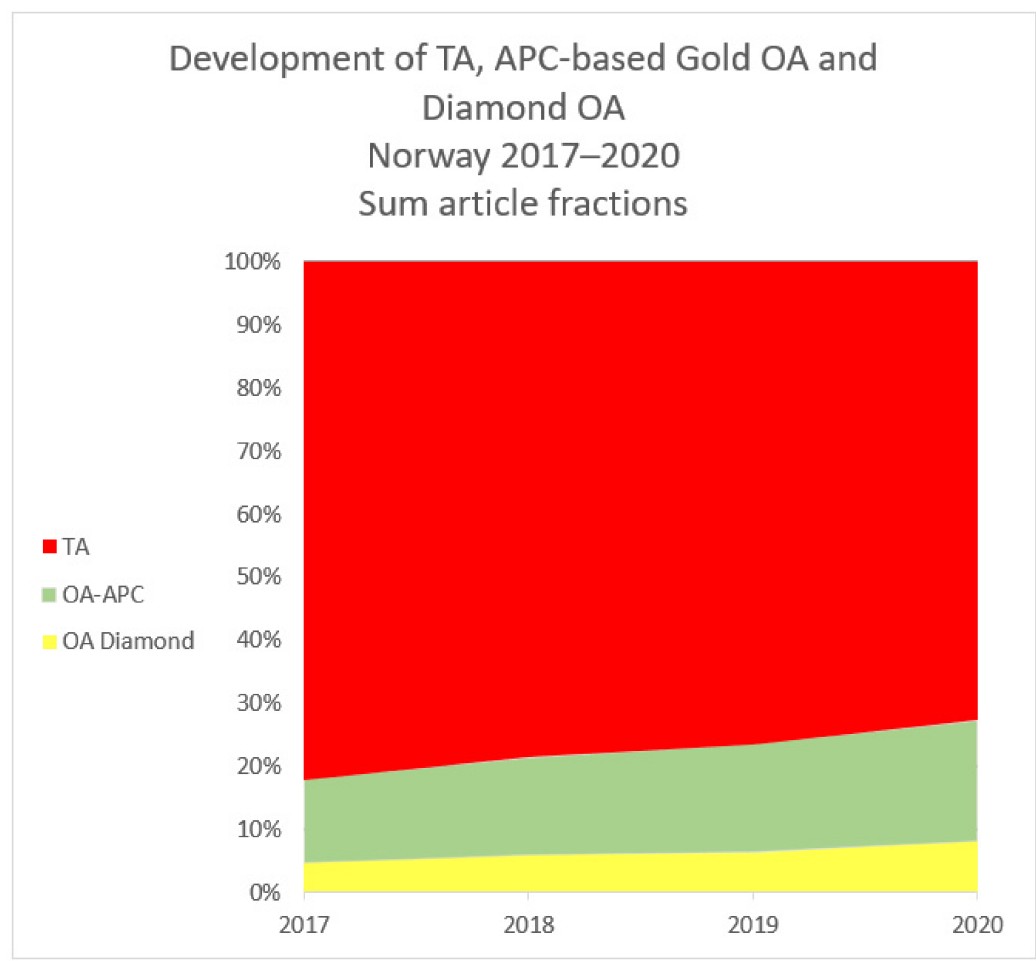

**Figure 1.** Development of Norwegian publishing 2017–2020 under various models.

From Table 1 we see that TA decreased from 82% to 73%, OA-APC grew from 13% to 19%, while Diamond OA grew from 5% to 8%.

**Table 1.** Development of Norwegian publishing 2017–2020 under various models, sum of article fractions.

| Model | 2017 | 2018 | 2019 | 2020 |
|---|---|---|---|---|
| TA | 10,582 | 10,180 | 10,533 | 10,526 |
| OA-APC | 1677 | 1997 | 2361 | 2770 |
| OA Diamond | 618 | 770 | 880 | 1177 |
| Sum all models | 12,877 | 12,947 | 13,774 | 14,473 |

TA is the most common mode of publishing, and the volume is relatively constant over the years studied, even if the share of the total is declining. We know that part of this is OA, namely Hybrid OA. In the HE sector, Hybrid has grown from 7.6% of all published articles in 2017 to 14.6% in 2019 to 24.8% in 2020 (percentages based on raw numbers in

Figure 3.6 in Haugen, et al. [11]). This means the actual percentage of Gold OA in Norway (the sum of APC-based OA in OA journals, Diamond OA, and Hybrid OA) in 2020 was probably higher than 50%. UNIT—The Norwegian Directorate for ICT and Joint Services in Higher Education and Research—is, amongst other things, saddled with the responsibility to negotiate Read&Publish deals with major publishers on behalf of the HE institutions, with a view to ensure as much OA as possible. In 2019, the first major such deals with Elsevier, Springer, Wiley, and Taylor & Francis were agreed and started operating, with new deals commencing in 2020 (Haugen, Holme, Håvik, Lie, Landøy, Osland, Rutledal, Ørnes and Pedersen [11]). Some purely OA deals have also been made. The growth in Hybrid from 2018 owes very much to these deals. OA in OA journals was generally well financed through institutional publication funds, so the OA deals only made this work more efficiently and should not in themselves create strong growth in OA-APC. A fear was that the deals would give authors easier access to Hybrid and that that could cannibalize on OA in OA journals, but that fear seems to have been unfounded as APC-based non-Hybrid OA has also grown during the last years.

### 3.2. Conversion of TA Journals to Diamond OA

That TA journals convert to OA, even Diamond OA, is part of the development of the journal market, and there are numerous examples of this. What needs special treatment in the case of Norway is a support mechanism called NÅHST (Norwegian Open Humanities and Social Sciences Journals) where the Ministry of Education and Research, the Norwegian Research Council, and a number of HE institutions joined together to finance the conversion of a number of HSS journals publishing mainly in Norwegian from a TA to a Diamond OA model. This included many of the older and more prestigious HSS journals in Norway. Part of the increase in the uptake of Diamond OA publishing in Norway is probably due to this mechanism. In the period 2018–2020, 25 journals were supported through NÅHST [12]. This mechanism replaced a support mechanism from the Norwegian Research Council for OA journals in 2017, but the journals covered by the old and the new mechanisms were not necessarily the same.

In Table 2 we find the number of articles published in OA Diamond journals without NÅHST support (the "No" column to the left) and the number of articles in such journals that now receive NÅHST support under the "Yes" heading. The latter numbers are divided into articles actually published as OA Diamond and articles published as TA articles before the journal converted to OA, with a sum column ("Sum Yes") to the right. Below, the same numbers are expressed as percentages of the total article production over all publication forms (OA, OA Diamond, TA) in the year in question. We see that from 2020, all articles in these journals supported by NÅHST were published OA.

We see that the volume of OA Diamond with NÅHST support increased, but this is due to the journals converting to OA, the total of articles published in these journals was roughly the same fraction of total publication all years ("Sum Yes" as a fraction of "Total production"). OA Diamond not supported by NÅHST grew from 3.7% to 5.9% of the total production over these four years.

This indicates that converting solid TA journals to Diamond OA, with good financial support, resulted in an increase in the uptake of OA, but it also shows that Diamond OA is of growing importance generally.

**Table 2.** OA Diamond with and without NÅHST support 2017–2020, sum of article fractions.

| Numbers | No | Yes | | | Total Norwegian Article Production over All Modes of Publication |
|---|---|---|---|---|---|
| | | **NÅHST Support** | | | |
| Year | OA Diamond | OA Diamond | TA | Sum Yes | |
| 2017 | 473 | 144 | 95 | 239 | 12,877 |
| 2018 | 539 | 232 | 52 | 284 | 12,947 |
| 2019 | 629 | 251 | 18 | 269 | 13,774 |
| 2020 | 860 | 317 | | 317 | 14,473 |
| Percentages | | | | | |
| 2017 | 3.7 % | 1.1 % | 0.7 % | 1.9 % | |
| 2018 | 4.2 % | 1.8 % | 0.4 % | 2.2 % | |
| 2019 | 4.6 % | 1.8 % | 0.1 % | 2.0 % | |
| 2020 | 5.9 % | 2.2 % | 0.0 % | 2.2 % | |

### 3.3. The Importance of Diamond OA in Main Scholarly Fields

In the database underlying Cristin, each journal is assigned to one scholarly (sub-) field. Some journals are difficult to place in a specific field, they are often placed in a top-level field. Assigning only one field to a journal creates problems for journals publishing over a wide field or in the intersection of two or more fields. This notwithstanding, the numbers should give us some picture of the importance of Diamond OA over scholarly fields, unless there is reason to believe Diamond OA journals are different from other journals when it comes to classifying which field they belong to.

### 3.3.1. Natural Sciences and Engineering

This was the largest scholarly field in terms of sum of article fractions for 2020. As seen in Table 3, TA was obviously dominant—nearly 80%, APC-based OA was important, OA Diamond was a small—but possibly growing—fraction of what was published.

**Table 3.** Publishing models in natural sciences and engineering 2019–2020, sum of article fractions.

| Sum Fractions | OA Diamond | OA-APC | TA | Grand Total |
|---|---|---|---|---|
| 2019 | 169 | 951 | 4612 | 5732 |
| 2020 | 185 | 1104 | 4586 | 5875 |
| Percentages | | | | |
| 2019 | 2.9 % | 16.6 % | 80.5 % | |
| 2020 | 3.1 % | 18.8 % | 78.1 % | |

### 3.3.2. Health Sciences

Health sciences is the second largest scholarly field.

As shown in Table 4, TA was dominant here, too—but was less than two thirds, declining from 69% in 2019 to 65% in 2020. The reduction in TA was almost entirely replaced by an increase in APC-based OA, though there was some growth in Diamond OA.

**Table 4.** Publishing models in health sciences 2019–2020, sum of article fractions.

| Sum Fractions | OA Diamond | OA-APC | TA | Grand Total |
|---|---|---|---|---|
| 2019 | 76 | 1112 | 2683 | 3872 |
| 2020 | 145 | 1267 | 2645 | 4057 |
| Percentages | | | | |
| 2019 | 2.0% | 28.7% | 69.3% | |
| 2020 | 3.6% | 31.2% | 65.2% | |

### 3.3.3. Social Sciences

This field is somewhat smaller than health sciences measured in sum of article fractions.

As seen from Table 5, the picture here is markedly different from natural sciences and engineering, and health sciences. TA was dominant and declining in relative share, APC-based OA smaller though growing, and Diamond OA was larger than APC-based and growing, both in numbers and percentage.

**Table 5.** Publishing models in social sciences 2019–2020, sum of article fractions.

| Sum Fractions | OA Diamond | OA-APC | TA | Grand Total |
|---|---|---|---|---|
| 2019 | 382 | 238 | 2185 | 2804 |
| 2020 | 526 | 324 | 2244 | 3094 |
| Percentages | | | | |
| 2019 | 13.6% | 8.5% | 77.9% | |
| 2020 | 17.0% | 10.5% | 72.5% | |

### 3.3.4. Humanities

The last and smallest of the four fields is humanities.

Here, too, TA was the major form as shown in Table 6, but declining in share of output, APC-based OA was a small part though growing, and Diamond OA was nearly a quarter of all publishing and growing strongly.

**Table 6.** Publishing models in humanities 2019–2020, sum of article fractions.

| Sum Fractions | OA Diamond | OA-APC | TA | Grand Total |
|---|---|---|---|---|
| 2019 | 253 | 60 | 1052 | 1365 |
| 2020 | 322 | 75 | 1051 | 1447 |
| Percentages | | | | |
| 2019 | 18.5 % | 4.4 % | 77.1 % | |
| 2020 | 22.2 % | 5.2 % | 72.6 % | |

The strong growth of Diamond OA in social sciences and humanities could partially be explained as an effect of NÅHST support, but the main effect of that came earlier, from 2017 to 2018, though it partially explains the relatively high share of Diamond OA in these fields.

### 3.3.5. Summing Up

From the numbers presented here, one should be able to conclude that

(a)    toll access is dominant in all fields, but also declining as a percentage of output in all fields.

(b)    APC-based OA is growing and is a major factor in health sciences.

(c) Diamond OA is small in natural sciences and engineering and health sciences, but growing somewhat—while being an important way of publishing both in social sciences and humanities and growing in importance.

### 3.3.6. Some Notes on Scholarly Subfields

Our raw data also contain information about the scholarly sub-field of the journals used. A detailed discussion of this would take much space, but some special cases can be noted. Sub-fields where the total production 2019–2020 was less than 100 articles are ignored as small numbers easily vary with no underlying cause but chance.

In health sciences, general medicine stands out with less than 50% TA—37% TA, 47% APC-based, and 16% Diamond OA. A closer look at the last number reveals this is mainly due to the Journal of the Norwegian Medical Association having become Diamond OA. A point about this journal is that while the HE sector is the major sector when it comes to volume of publications, the HE sector only provides about 1/3 of the content in this journal. This journal is obviously relatively more important for the health sector (total author fractions 2019 67.19, 2020 64.06—while corresponding numbers for the HE sector are 23.22 and 22.5, respectively; total numbers from the data set used, HE sector numbers from [13]).

One should note that in some fields connected to health professions, there are society journals that give free access to their content, the content is freely available on the Internet. However, as many of these journals have not taken the last steps towards becoming truly OA, they are currently what is termed by some Bronze OA, i.e., giving free access to content but have no reuse licensing. They are here listed as TA, as they are not in DOAJ and they do not currently qualify for a listing there. Taking the final steps to make them OA should not create problems for their income side as they already make content freely available, but it would make them more visible and acceptable as publishing venues for authors under a contractual OA obligation. This should indicate that even in the health sciences, the dependence on a Diamond-like model is higher than the numbers here show.

In natural sciences and engineering, physics has 13% Diamond OA; this seems mainly to be due to some high energy physics (HEP) journals—this could be the result of SCOAP3, the consortium led by CERN to promote OA output for HEP. Environmental technology and industrial ecology has 16% Diamond OA, this seems to result from conference proceedings published as Diamond OA.

In social sciences, gender studies (38% Diamond) and interdisciplinary social sciences (30% Diamond) stand out. On the other end of the scale, we find geography with 1% Diamond, economics with 2% Diamond, and development studies with 5% Diamond and 0% APC-based OA. One would have thought development studies would have found it beneficial to have their content easily available in poorer countries.

In the humanities, archaeology and conservation with about 8% Diamond OA is at the low end of the scale, and history and literature with 34% and 31% Diamond OA at the high end.

### 3.3.7. Nationality and Language

Diamond OA is an international phenomenon, but many such journals have a local focus, and they often publish in local languages. A close look at language and country of publisher shows that 58% of the Diamond OA content is published in Norwegian journals; namely 74% in humanities, 67% in social sciences, 51% in health sciences, and 12% in natural sciences and engineering. In health sciences, the journals owned by Norwegian professional societies explain the relatively high percentage of publishing in national journals for such an internationally oriented scholarly field, while humanities and social sciences journals often publish scholarship with a more local focus and for a local language audience. The low percentage for natural sciences and engineering is commensurate with an international scholarly field with an international readership.

The data from HK-dir only contain information on publishing language for some of the journals, but DOAJ has language information for most journals. However, this is information on the languages for which the journal accepts manuscripts. Many journals accept more than one language, and one cannot know which language has been used for content, generally. If one assumes that Norwegian authors prefer to publish in Norwegian when this is an option, or that other Nordic languages are used if possible [1], Table 7 shows these numbers for language use:

**Table 7.** Language use across scholarly fields in Diamond OA journals 2019–2020.

| Sum of Article Fractions 2019–2020 | Language | | | |
|---|---|---|---|---|
| Scholarly Field | English | Nordic | Other | Grand Total |
| Health Sciences | 94 | 126 | 1 | 221 |
| Humanities | 166 | 396 | 13 | 575 |
| Natural Sciences and Engineering | 353 | | 1 | 354 |
| Social Science | 343 | 565 | | 908 |
| Grand Total | 956 | 1086 | 14 | 2057 |

A manual check on journal titles reveals that this assigning of language to content generally holds true. Most content marked as Nordic is published in journals where English is not really an option.

We see that there is a majority of Nordic language content except in natural sciences and engineering where Nordic languages are not used at all, only English. "Other" turns out to be mainly Spanish, though there are also other Romance languages and Russian.

In APC-based publishing, most journals are monolingually English. The above table corresponds well with the finding in [4] that Diamond OA journals generally were more linguistically diverse than APC-based OA journals.

There has been recent debate in Norway about the importance of publishing in Norwegian, see e.g., [14]. The numbers here indicate Diamond OA journals stand out as a possible answer to the political pressure to publish in the national language.

3.3.8. The Norwegian "Points" System

Scholarly output from Norwegian scholars is counted and reported on, and for many institutions, the publication activities count towards the financing of the author's institution. This is based on a system where journals and publishers are accredited in order to count towards giving points to authors. A standard journal article on level 1 gives 1 point, to be divided between authors and institutions according to a set of rules. Some journals are singled out on level 2 as more important and valuable than others, publishing in such a journal gives 3 points to divide between the authors. Only a limited set of journals can be "elevated" to level 2. They must not contain more than 20% of the global production of articles in their field. There are lively discussions on which journals to list at level 1 and 2, this impacts authors a lot. A publication "point" is worth about 20,000 NOK (roughly 2000 EUR) to the author's institution, so there is considerable pressure to publish on level 2 for purely economic reasons. A good description of this is given in [5].

Table 8 shows how the publishing volumes for 2019 and 2020 were distributed over access types and levels in the Norwegian system. It seems obvious that this system clearly benefits publishing in TA journals, where more than 25% was published on level 2, while APC-based OA had 13% at level 2 and Diamond OA only 6%. It is interesting to note the growth in Diamond OA over the last few years despite the low fraction of top-tiered Diamond OA journals in the Norwegian system.

**Table 8.** Publishing 2019–2020 OA type and level.

| | Total of Article Fractions | | Relative Percentages | | Total |
|---|---|---|---|---|---|
| | Level 1 | Level 2 | Level 1 | Level 2 | |
| OA Diamond | 1935 | 121 | 94.1 % | 5.9 % | 2057 |
| OA-APC | 4461 | 671 | 86.9 % | 13.1 % | 5131 |
| TA | 15,660 | 5399 | 74.4 % | 25.6 % | 21,059 |
| Grand Total | 22,056 | 6191 | 78.1 % | 21.9 % | 28,247 |

This Diamond OA disadvantage is not a policy in itself, but a result of a number of circumstances. One is that level 2 journals should have a high standing and a journal in a small language (if we include the whole Nordic language area, we are talking of a population of 20–22 million) will have relatively few readers, hence fewer citations and a correspondingly low standing internationally. That does not mean the journal content is of low quality. There are also incentives in the financing system to publish with international co-authors, this will favor international journals. In some fields, such national Diamond OA journals will be quite general, e.g., the Journal of the Norwegian Medical Association; this is generally a handicap in the quest for level 2 status, which favors specialized journals.

## 4. Discussion

My assumption before this study was that Norway would show the same distribution of Diamond OA over scholarly fields as in the OA Diamond study, while the share of Diamond OA would be lower because the more commercial forms of publishing are well funded, increasingly so with the deals with major publishers enabling easy access to hybrid publishing. Moreover, Norwegian scholars have to be able to communicate well in English—there is little need for local language journals to accommodate authors unable to publish in English. However, local language journals are important in order to reach out beyond academia.

The numbers seem to indicate that

- OA in general is of growing importance, more so as Hybrid OA is not showing up in the numbers but is hidden in the TA numbers.
- APC-based OA in OA journals is growing.
- Diamond OA is growing.
- Diamond OA is very important, and increasingly so, in humanities and social sciences.
- Diamond OA seems to be important for scholarship in the national language.

The relative importance of Diamond OA is actually on the same level as the global level in the OA diamond study, with 8% compared to 8–9% as the share of Diamond OA of all publications—this seems surprisingly high. On the other hand, APC-based OA has a much larger share—19%—in Norway than was found globally in the OA Diamond Study, where this was estimated to be 10–11%. Consequently, Diamond OA has a smaller share of the OA published in OA journals in Norway than globally.

The OA Diamond report does not contain information on the relative distribution of Diamond OA and APC-based OA across scholarly fields. Such information was studied in the OA Diamond project; Table 9 gives the broad picture:

**Table 9.** Relative shares of Diamond and APC-based OA from the OA Diamond project.

| Scholarly Field | Diamond OA | APC-Based OA |
|---|---|---|
| -    Humanities | 81% | 19% |
| -    Medicine | 30% | 70% |
| -    Sciences | 31% | 69% |
| Total | 44% | 56% |

Table 10 shows that in the present study, we find this distribution between Diamond OA and APC-based OA over scholarly fields in Norway 2019–2020:

**Table 10.** Relative shares of Diamond and APC-based OA in Norway 2019–2020.

| Scholarly Field | Diamond OA | APC-Based OA |
|---|---|---|
| -    Humanities | 68% | 32% |
| -    Medicine | 8% | 92% |
| -    Sciences | 15% | 85% |
| Total | 29% | 71% |

The definitions of scholarly fields may not necessarily be the same in the two sets of data, and while the OA Diamond data count articles in journals, the present data sum up article fractions for Norwegian authors. The OA Diamond data are a sum of average number of articles in journals over three years 2017–2019.

We see that Diamond OA as a share of OA is markedly lower in Norway than globally. This varies by scholarly fields, where it looks like medicine is the field where Diamond OA is the weakest compared to the OA Diamond study's finding. This could be due to a combination of a lack of high-ranking journals for publishing in Scandinavian languages in the field of medicine and a well-funded APC-based Gold OA option.

Norwegian seems to be a relatively more important publishing language in Diamond OA journals than the OA Diamond study results seem to indicate. This could possibly in part be a result of HSS being relatively more important for Norwegian authors publishing Diamond than in the global landscape.

The growth of Diamond OA is a clear sign that this kind of OA is in demand among authors—it is growing despite the existence of increasingly better funded competitors such as APC-based OA and Hybrid OA. This is also despite the Norwegian financing system actually favoring the TA and APC-based/Hybrid OA models through the choice of journals for the top tier.

The likelihood that quite a large number of OA journals globally is not listed in DOAJ points to a need for national-level organizations to offer help in preparing journals for applying to register in DOAJ and with the application process itself. A more complete list of OA journals in DOAJ might show Diamond OA to be more important than current numbers show, and would increase the value of these journals to their authors.

Having in mind that Diamond OA is the publishing model where scholars and their institutions are most in control of the publishing activities, the importance of Diamond

OA for many scholars should indicate a need for institutions and funders to prioritize this model in their funding considerations.

**Funding:** This research received no external funding.

**Data Availability Statement:** The raw data that these analyses are based on are made available in UiT Open Research Data https://doi.org/10.18710/LHRTP9 (deposited and accessed on 2 March 2022) under the license decided by the original data providers.

**Acknowledgments:** Thanks to Vidar Røeggen of Universities Norway and Per Pippin Aspaas of UiT The Arctic University of Norway for reading through and commenting on earlier versions of this MS, and to Peter Millington, retired SHERPA Services Development Officer, University of Nottingham, for helping with the language. All remaining errors are the author's. Thanks also to The Norwegian Directorate for Higher Education and Skills for providing a specially prepared data set for this analysis.

**Conflicts of Interest:** The author is a member of *Publications'* Editorial Board.

## Notes

1    (Written) Norwegian, Danish, and Swedish are mutually intelligible and academics with one of these as mother tongue are expected to read the other two languages without problems. The same goes for students.

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
