# Peer review of "Diamond Open Access in Norway 2017–2020"

_publications, doi:10.3390/publications10010013_

Round 1

Reviewer 1 Report

The idea suggested in the abstract to develop national- or regional-level case studies to complement the Mar'2021 "OA Diamond Journals Study" commissioned by cOAlition S is a very valuable one. Furthermore, the manuscript looks into the appropriate indicators within the Norwegian publishing landscape whose analysis could lead to a useful case study, one that could be potentially replicated for other countries or regions.

However, the paper reads like an early draft and has numerous issues, the most severe of which is the fact that no comparison is provided in a rather short "Item 4. Discussion" (p. 10) between the results of this Norwegian case study and the findings of the global Diamond OA report listed as the first reference in the bibliography.

Some context to this global Diamond OA report should ideally also be provided in the introduction -- its absence seems to assume that every reader is aware of this report and its findings besides the circumstances that led to its publication. This may be the case at this point in time -- less than a year since the report was published -- but probably not after an additional year of two have gone by. If the main findings were summarised in the introduction, the discussion at the end of the paper could look into whether these findings are confirmed by the case study

Some of the issues in the manuscript referred to above are:

  • The reading of the paper would much benefit from using bold font for sub-section headings within the major sections -- the current impression is somewhat 'drafty' and makes the structure of the paper occasionally hard to follow
  • English needs to be improved: "This is, however, numbers on a global scale" (line 10 in abstract) or "data are on the journal, not on the article level" (line 24) are but a couple of examples
  • The stated inability to measure Hybrid OA is a major flaw in the study. The reference "In the HE sector, Hybrid has grown from 7,6% of all published articles in 118 2017, to 14,6% in 2019 to 24,8% in 2020 (Percentages based on raw numbers in figure 3.6, 119 Haugen, Holme, Håvik, Lie, Landøy, Osland, Rutledal, Ørnes and Pedersen [3])" is only on p. 4 and should perhaps be mentioned earlier (the reference is actually there on line 26 but not the figures)
  • The expression (line 34) "Norway has a centralized CRIS (Cristin) where all Norwegian research from the Higher Education (HE), Health and Institute sectors is registered" is confusing: does it mean Health and Research Institute sectors? The following quotation could be a useful replacement: "Norway is one of  a  few  countries  that  has  a  fully  integrated  non-commercial  CRIS  system  at  the  national  level. Cristin  (The  Current  Research  Information  System  in  Norway;  cristin.no)  is  a  shared  system  for  all  research  organizations  in  the  public  sector:  universities,  university  colleges,  university  hospitals  and  independent  research  institutes" (source: G Sivertsen (2018) "The Norwegian Model in Norway". Journal of Data and Information Science 3(4):2-18, DOI:10.2478/jdis-2018-0017). This also provides the URL for Cristin, which is nowhere to be found on this paper
  • The explanation on the 'Norwegian model' would benefit from the inclusion to a citation to G Sivertsen's paper above, or to this other one: J Pölönen, R Guns, E Kulczycki, G Sivertsen, TCE Engels (2021). National lists of scholarly publication channels: An overview and recommendations for their construction and maintenance. Journal of Data and Information Science, 6(1), 50–86. DOI: 10.2478/jdis-2021-0004
  • Reference 6 from page 5 is lacking a URL, https://www.openaccess.no/rapport-evaluering-nahst.pdf. This is a somewhat unique publicly-funded mechanism to support the transition to OA that readers in other countries may want to know more about even if the report is in Norwegian
  • Captions for table 2 (and successive ones) do not explain if the figures are for total number of articles or journal titles
  • Percentages in table 4 have been copied from table 3 and are mathematically wrong. Also, the explanatory text below the table suggests that figures are available from 2017 to 2020, while only 2019 and 2020 are shown on the table
  • Line 227 "what is by some termed Bronze OA" belongs in the methods section together with the considerations on Hybrid OA that have indeed been included there at the start
  • Percentages on table 8 surprisingly reflect the distribution of the various publishing models across levels but not the percentage of each of the publishing models with regard to the totals for levels 1 and 2. Diamond OA accounts for 8,8% of level 1 titles and 1,9% of level 2 titles. The absence of these figures might suggest that figures that do not show positive results for the model the article is promoting are simply dismissed
  • The fact that the author of this paper is one of the co-authors for the global Diamond OA report might allow some qualitative info to be provided on how many replies to the global survey were received from Norway or what the report hints at for Norway in terms of the technical publishing standards (or other aspects such as voluntary work) from a basic analysis of the appropriate subset of the replies  

Author Response

Thank you to the reviewers for comments, they have helped me make this MS much better. Both by pointing to errors or content easily misunderstood, to structural problems and to content missing. Especially the need to actually compare the Norwegian numbers to the OA Diamond study numbers was very important.

The comments also have shown a need to tone down my treatment of other forms of OA than Diamond OA, which is what was intended to be the focus – hopefully, this comes clearer through now.

I also discovered that I had used knowledge I had accumulated during the number crunching for the Diamond OA report as if it was published, but it turned out some important facts hadn’t been part of the report – there is a limit to how many numbers and tables you can have in a report before the readership dwindles. So some results of that study is presented for the first time in the Discussion section.

I have commented on Reviewer 1’s comments in the word document containing the comments, my commenting is indented. I have used the commenting function in the PDF to comment on Reviewer 2’s comments.

For easy reading I have, in addition to the reworked MS with changes tracked, also uploaded a new version with all changes incorporated for new comments. With so many changes the Track Changes option makes a document difficult to read.

Reviewer 2 Report

Dear author, thanks for your manuscript. Habe a look at my notes in your text. Best wishes for your publication.

Author Response

Replies in the comments in the PDF file.

Thank you to the reviewers for comments, they have helped me make this MS much better. Both by pointing to errors or content easily misunderstood, to structural problems and to content missing. Especially the need to actually compare the Norwegian numbers to the OA Diamond study numbers was very important.

The comments also have shown a need to tone down my treatment of other forms of OA than Diamond OA, which is what was intended to be the focus – hopefully, this comes clearer through now.

I also discovered that I had used knowledge I had accumulated during the number crunching for the Diamond OA report as if it was published, but it turned out some important facts hadn’t been part of the report – there is a limit to how many numbers and tables you can have in a report before the readership dwindles. So some results of that study is presented for the first time in the Discussion section.

I have commented on Reviewer 1’s comments in the word document containing the comments, my commenting is indented. I have used the commenting function in the PDF to comment on Reviewer 2’s comments.

For easy reading I have, in addition to the reworked MS with changes tracked, also uploaded a new version with all changes incorporated for new comments. With so many changes the Track Changes option makes a document difficult to read.

Round 2

Reviewer 1 Report

Thanks for having efficiently addressed in this updated ms the comments in the peer-review report. The article is now much improved in my view and all key considerations have been taken into account. No further changes are recommended beyond a succinct English style review.

Author Response

Thanks for good comments and questions, the MS has been to a native speaker who is an academic with an OA background, his comments have been worked into the final version submitted.